# Cost–Benefit Evaluation of Walnut and Jujube Orchards under Fruit Tree–Crop Intercropping Conditions in Southern Xinjiang

Jingyu Jin [1,2,3,4], Jie Bai [1,2,*], Anming Bao [1,2], Hongwei Han [5], Junli Li [1,2], Cun Chang [1,2] and Jiayu Bao [6]

1   State Key Laboratory of Desert and Oasis Ecology, Key Laboratory of Ecological Safety and Sustainable Development in Arid Lands, Xinjiang Institute of Ecology and Geography, Chinese Academy of Sciences, Urumqi 830011, China; 18160212966@163.com (J.J.); baoam@ms.xjb.ac.cn (A.B.); lijl@ms.xjb.ac.cn (J.L.); changcun@ms.xjb.ac.cn (C.C.)
2   Key Laboratory of RS & GIS Application Xinjiang, Urumqi 830011, China
3   University of Chinese Academy of Sciences, Beijing 100049, China
4   Survey and Mapping Achievement Center of Xinjiang, Urumqi 830002, China
5   Research Institute of Economic Forestry, Xinjiang Academy of Forestry Sciences, Urumqi 830063, China; ecoforest@126.com
6   Faculty of Land Resources Engineering, Kunming University of Science and Technology, Kunming 650093, China; christina_baob@163.com
*   Correspondence: baijie@ms.xjb.ac.cn; Tel.: +86-136-6996-7925

**Abstract:** Fruit planting plays an essential role in achieving zero hunger, ensuring food security, and increasing the income of local people. As one of China's principal fruit-planting areas, southern Xinjiang possesses more than 80% of the total fruit-planting areas of Xinjiang. However, the spatial distribution, extent, and types of fruit trees remain unknown, generating uncertainty in calculating the economic benefits of orchards in this area. In this paper, we used walnut and jujube orchards under fruit tree–crop intercropping conditions in the Hotan Oasis in southern Xinjiang as the research object. Walnut and jujube orchards were precisely characterized using object-oriented and decision tree classification methods. Then, the economic benefits for farmers were estimated based on field surveys and statistical data. From 2003 to 2020, the area of jujube orchards rapidly increased from $1.91 \times 10^3$ ha to $33.59 \times 10^3$ ha, while that of walnut orchards steadily increased from $18.05 \times 10^3$ ha to $40.24 \times 10^3$ ha. The expansion areas of walnut orchards were mainly transformed from other orchards in the oasis, while the expansion areas of jujube orchards mainly originated from newly cultivated land in the desert. The increases in areas and yields largely offset the increase in planting costs and the decrease in purchase prices, resulting in an increasing trend in the total net income of the orchards. From 2003 to 2020, the total net income of walnut orchards increased by 68.96%, and the total net income of jujube orchards significantly increased by 23.37 times. However, the output/input ratios of walnut orchards under intercropping conditions were approximately two times higher than those of jujube orchards under monocropping conditions. The increase in investment slightly offset the decreases in yields and purchase prices, resulting in increases of 5.24% and 18.64% in the output/input ratios of walnuts and jujubes, respectively, in orchards exhibiting standardized cultivation. In the future, it is necessary to significantly expand the area of orchards involving standardized cultivation and improve the yield and quality of fruits, thereby increasing the yields and income levels of local farmers.

**Keywords:** walnut orchards; jujube orchards; intercropping; decision tree classification; multitemporal remote sensing; output/input ratios; southern Xinjiang



## 1. Introduction

Due to the intensifying extreme climate conditions and growing global population, food insecurity has become an increasing concern on the global scale, while it also poses a considerable challenge to several of the Sustainable Development Goals (e.g., Sustainable

Development Goal 2: zero hunger) in many developing countries [1–4]. Fruit products (including fresh and dried fruits) are an important part of food, which can supplement the shortage of grain products and enhance the nutrient balance of the dietary structure of humans. In addition, producing, processing, and marketing fruit products can improve the living conditions of smallholder farmers, increase employment opportunities, and promote local economic development [5]. Producing, processing, and marketing fruit can also help to alleviate absolute poverty in remote rural areas and consolidate the achievements of poverty alleviation. Therefore, developing the fruit industry could greatly contribute to achieving zero hunger and ensuring food security.

As a significant fruit-producing region globally, China produces 270 million tons of fruit, accounting for nearly half of the global fruit production (FAO Database 2019). Xinjiang is a critical fruit-producing area in China, with 13% of the national fruit-planting area [6], and it is also one of the world's top six major fruit-producing regions. By 2020, the planting areas of fruits in Xinjiang had exceeded $14.67 \times 10^5$ ha, accounting for 65.77% of the agricultural land area (Xinjiang Statistical Yearbook). Fruit yields reached $12 \times 10^6$ tons, and the output value of fruits exceeded CNY $10 \times 10^9$ [7]. Due to its long history of cultivation, unique climatic conditions, and diverse ecological environment, southern Xinjiang has become the main fruit-planting area of Xinjiang, and its fruit-planting area constitutes more than 80% of the total fruit-planting area of Xinjiang. The fruit yields and output value of southern Xinjiang account for 76.93% and 83.79%, respectively, of those of the whole Xinjiang region. The income from fruit planting accounts for more than 60% of the income of local farmers, and the fruit industry is one of the most important economic industries for increasing the income of local farmers [8]. With the continuous increase in the areas and yields of fruit cultivation, the spatial information of the orchard distribution, extent, and type becomes increasingly unclear, yielding uncertainty in calculating the economic benefits of orchards in this region. Therefore, it is important to conduct accurate and real-time economic benefit evaluation studies of fruit trees to improve the quality and efficiency of the fruit industry, increase the income of farmers, consolidate the achievements of poverty alleviation, and realize rural revitalization in southern Xinjiang.

Accurate extraction of the spatial distribution and the extent and type of fruit trees constitute the basis of evaluating the economic benefits of orchards, which is important for fruit production. Most orchards are located in remote rural areas, with inconvenient transportation conditions, adverse topography, and irregular coverage areas. The traditional survey methods for orchards (including household surveys, statistical reporting, and sampling surveys) are subjective, time-consuming, costly, and lack spatial information, which inevitably influences the accuracy of survey results. With the advantages of objectivity, high timeliness, comprehensive coverage, and low cost, remote sensing technology has been widely used for orchard mapping and monitoring, such as fruit tree classification, area extraction, growth monitoring, and yield estimation in large regions [9]. Low- and medium-spatial resolution satellite images, which provide the advantage of a high temporal resolution, are usually used to construct time series of vegetation index curves [10] and are very suitable for extracting orchards with large-scale areas and homogeneous fruit tree fields. Based on time series analysis of MODIS, Landsat, HJ–CCD, and Sentinel satellite images, the areas of apple, citrus, jujube, sugarcane, cashew, and other orchards have been extracted using decision tree classification approaches [11], object-oriented methods [12], and machine learning models. These studies have shown that high-temporal resolution imagery is critical to accurate planting area delineation [13,14]. The Chinese high-resolution Gaofen (GF) series satellites with a submeter-level spatial resolution have become an important high-resolution data source for accurately depicting orchard field boundaries. Many researchers have used object-oriented methods or machine learning models to extract the spatial distribution and extent of orchards based on single GF satellites considering spectral, textural, and spatial features [15–18]. However, high-resolution satellite images exhibit low spectral and temporal resolutions, and they could suffer the phenomenon of different objects with the same spectrum. Therefore, comprehensively combining multispectral

and high-resolution satellite images for extracting orchards has become a research hotspot. Most studies on orchard mapping have focused on mono-planting areas rather than intercropping areas. As the patches of intercropping areas exhibit high complexity, spectral mixture, and texture aliasing are typical in remote sensing images and are easily affected by understory crops. Therefore, it is more difficult to accurately extract intercropping fruit trees using mono-temporal remote sensing images.

Fruit planting greatly depends on climate, soil, topography, water resources, and other natural conditions. There are many studies on the suitability analysis of fruit planting based on environmental and socioeconomic conditions [19,20]. Planting fruit trees in suitable areas is more likely to produce higher yields and economic benefits. Scientists have conducted energy and economic cost–benefit analyses of fruit trees, and they have also found that the net income derived from fruit trees under intercropping conditions is considerably higher than that under monoculture conditions. With the development of intensive agriculture, standardized cultivation in orchards with neat rows and columns, concentrated continuous areas, excellent varieties, modern agricultural machinery, and standard management techniques are increasingly popularized in fruit planting. Compared with traditional planting modes, it produces higher fruit yields with high quality and thus higher economic benefits. Therefore, this mode should also be considered in economic analysis. A reasonable and objective assessment of economic benefits requires accurate planting area determination and classification for orchards. However, the required data are usually obtained from statistical data or household surveys, so they exhibit a specific time lag and lack spatial information. The distribution, extent, and types of orchards involving standardized cultivation extracted from satellite images can improve the accuracy and timeliness of economic benefit evaluation.

The objectives of this study were (a) to accurately extract the area of walnut and jujube orchards under fruit tree–crop intercropping conditions based on high-resolution and multitemporal satellite images and (b) to evaluate the economic benefits of orchards, including standardized cultivation in orchards in 2003 and 2020.

## 2. Materials and Methods

### 2.1. Study Area

Located in the southern part of Xinjiang in north-western China, the Hotan Oasis covers an area of 3110 km$^2$ and encompasses the administrative districts of Hotan city, Hotan County, Moyu County, and Luopu County. The Hotan Oasis is mainly located in the diluvial plain between the Yurongkash River and the Karakash River, which is dominated by an artificial irrigation oasis. It belongs to the warm temperate, arid desert climate zone, with an average annual temperature of 11.3 °C and an average annual precipitation of 36.5 mm. It also has abundant solar and thermal resources, with a frost-free period of 170~201 d, an annual total radiation ranging from 138.1 kcal/cm$^2$ to 151.5 kcal/cm$^2$, and an annual accumulated temperature ($\geq$10 °C) of 4200 °C. The soil here is sandy with favourable permeability and moderate fertility, and the organic matter content in the 0~100 cm soil depth range is 6.3 g/kg. The Hotan Oasis is associated with more than 2000 years of fruit tree cultivation, and the characteristic fruit trees are walnuts, jujubes, grapes, and apricots. By 2019, the area of fruit trees reached $14.6 \times 10^5$ ha, with more than half the area devoted to walnuts and jujubes ($11.07 \times 10^4$ ha and $37 \times 10^3$ ha, respectively). As arable land resources are limited, this area mainly exhibits a fruit tree–crop intercropping pattern, and the main crops intercropped underneath fruit trees are winter wheat, corn, cotton, vegetation, and forage grass (Figure 1). Fruit trees are also planted in traditional courtyards or along roads, while standardized cultivation in orchards occurs in very few areas. The croplands in this region are densely fragmented and intertwined with houses, orchards, roads, trees, and other ground objects, rendering fruit tree extraction very difficult.

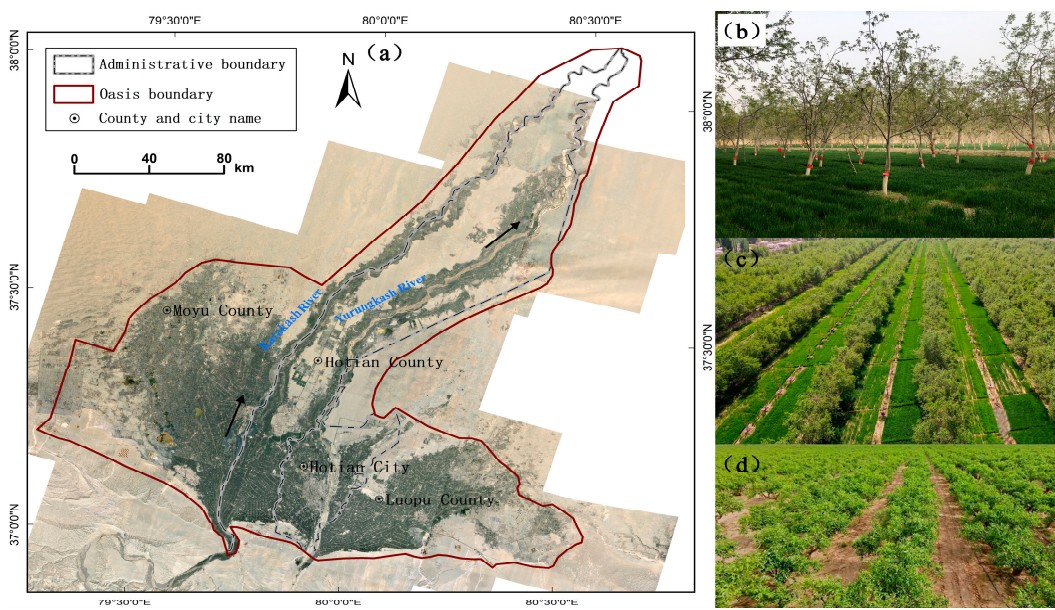

**Figure 1.** Locations of the Hotan Oasis in the southern part of Xinjiang, Northwest China (**a**); photograph of a traditional walnut orchard (**b**) and standardized cultivation of walnut orchards (**c**) and jujube orchards (**d**). Notes: The black arrows indicate the direction of the river.

*2.2. Data Source*

2.2.1. Remote Sensing Images

In this study, we used four satellite image types, providing data with different spectral and spatiotemporal resolutions (Table 1). High-resolution remote sensing images (WorldView-2 and Gaofen-2) were used to delimit patch boundaries. In contrast, multitemporal Landsat 5 Thematic Mapper (TM) and Sentinel-2 images were used to construct the normalized difference vegetation index (NDVI) time series. The data are described below (Table 1).

**Table 1.** Details of the remote sensing images in this study.

| Type of Remote Sensing Data | Satellite/Sensor | Spatial Resolution (m) | Acquisition Time | Amount |
|---|---|---|---|---|
| High-resolution data | WorldView-2 | 0.5 | June–September 2003 | 42 |
| Multi-temporal data | Landsat 5 TM | 30 | March–October 2003 | 82 |
| High-resolution data | GF-2 PMS | 1 | July–September 2020 | 35 |
| Multi-temporal data | Sentinel-2 MSI | 10 | March–October 2020 | 44 |

A total of 42 WorldView-2 (WV-2) images with a cloud coverage rate lower than 20% ranging from July to September 2003 were obtained from commercial procurement. The images encompassed eight multispectral bands at a 2.0 m spatial resolution and a panchromatic band with a spatial resolution of 0.5 m. A total of 35 Gaofen-2 (GF-2) images with a cloud coverage rate of less than 10% from July to September 2020 were downloaded from the China Center for Resources Satellite Data and Application (CRESDA, https://data.cresda.cn, accessed on 6 September 2022). The images contained four multispectral bands at a 3.2 m spatial resolution and one panchromatic band with a spatial resolution of 0.8 m. We used the Landsat 5 TM 30 m Collection 1, Tier 1, Surface Reflectance (C01/T1_SR) product derived from the Google Earth Engine (GEE) platform, which is a cloud-based platform providing a plethora of satellite images including Landsat sensor time series data from the United States Geological Survey (USGS, http://earthexplorer.usgs.gov, accessed on 15 October 2003). A total of 82 Landsat 5 TM images with a cloud coverage rate of less than 10% were distributed from March to October 2003. A total of 44 multispectral Sentinel-2 images (10 m spatial resolution) ranging from March to October 2003 were

selected and downloaded from the ESA data-sharing website (https://scihub.copernicus.eu/dhus/#/home, accessed on 18 October 2022). Sentinel-2 Atmospheric Reflective Top (L1C) products were used in this study, and these products were ortho-corrected and geometrically corrected on a fine scale.

2.2.2. Auxiliary Data

The administrative boundary of the county in 2018 at the 1:1,000,000 scale was obtained from the National Geographic Information Resources Catalogue Service System (https://www.webmap.cn/main.do?method=index, accessed on 18 October 2022). The ALOS PALSAR digital elevation model (DEM) with a spatial resolution of 12.5 m was downloaded from the Google Earth Engine (GEE, https://developers.google.com, accessed on 20 October 2022).

Socioeconomic data of the Hotan Oasis from 2003 to 2020 were obtained from the Statistical Yearbook and Government Bulletin, as listed in Table 2 below.

**Table 2.** Details of the socioeconomic data of the Hotan Oasis.

| Names | Period | Region | Source |
|---|---|---|---|
| Area of walnut/jujube orchards | 2003–2020 | | |
| Walnut/jujube yield | 2003–2020 | | |
| Output value of walnut/jujube cultivation | 2003–2020 | Hotan city, Hotan County, Moyu County, and Luopu County | Hotan Statistical Yearbook |
| Total output value of fruits | 2003–2020 | | |
| Total agricultural output value | 2003–2020 | | |
| Per capita net income of farmers | 2003–2020 | | Hotan Yearbook |
| Area of standardized cultivation in orchards | 2003–2020 | Hotan Prefecture | Hotan Yearbook and Hotan Statistical Yearbook |
| Purchase price of walnuts/jujubes per kg | 2003–2020 | China, Hotan Prefecture | https://wenku.baidu.com/view/1c0f16db240c844769eaee76.html; http://www.zao7.cn, https://www.chinabgao.com/jiage/hetao/, accessed on 22 October 2022 |

2.2.3. Field Data

A field survey was conducted in the Hotan Oasis in late April 2021, and 1052 effective sample points were collected. Among them, 482 were intercropping walnut sample points, 119 were monocropping walnut sample points, and 321 were jujube tree sample points. Of these, 19% served as training sample data and 81% served as accuracy validation data. At the same time, the residents of 28 townships and 167 villages in Hotan city, Moyu County, Hotan County, and Luopu County were interviewed and surveyed. Data were collected by combining on-site interviews and telephone questionnaires. To ensure the reliability of the data obtained, the questionnaires were adjusted and combined with pre-surveys, and 290 questionnaires were distributed, with an effective rate of 90%. Each questionnaire was divided into three parts: (1) basic information on orchard growers, including the number of labourers, planting years of the orchard, planting area of walnut and jujube orchards, yield and the output value of walnuts and jujubes per ha of orchard, input cost and income per ha of orchard, and per capita income of farmers; (2) input of the orchards, including the cost of irrigation, fertilization, pesticide, machinery, and hired labour; (3) output value of the orchards, including the purchased price of orchards and intercropping crops planted at the base of the fruit trees.

*2.3. Methods and Validation*

2.3.1. Overall Technical Scheme

In this study, the distribution and extent of walnut\jujube orchards in 2003/2020 were extracted using object-oriented and decision tree classification methods based on multi-source remote sensing images, and an economic benefit evaluation of these walnut/jujube

orchards was then performed based on the collected field survey data and statistical data. The data needed for the calculation of the output–input ratio are based on the field survey data from 2021. The socioeconomic data used for the trend analysis of the fruit industry from 2003 to 2020 were obtained from statistical yearbooks and government bulletins. The overall technical process can be described as follows: (1) based on high-resolution remote sensing images (WorldView-2 and Gaofen-2), the precise boundaries of orchards were extracted with a multiresolution segmentation algorithm and object-oriented classification methods. The NDVI time series derived from the multitemporal images (Landsat-5 TM and Sentinel-2) and phenology curves of fruit trees were analysed. Orchard patches were classified as walnut or jujube orchards using the decision tree classification method. Moreover, standardized cultivation areas of orchards in 2020 were extracted using geographical knowledge and interpretation. (2) A cost–benefit economic analysis of walnut and jujube orchards, including standardized cultivation in orchards, was conducted by integrating the field survey data and socioeconomic statistical data into the spatial information of orchards extracted from remote sensing images. The specific technical route is shown in Figure 2.

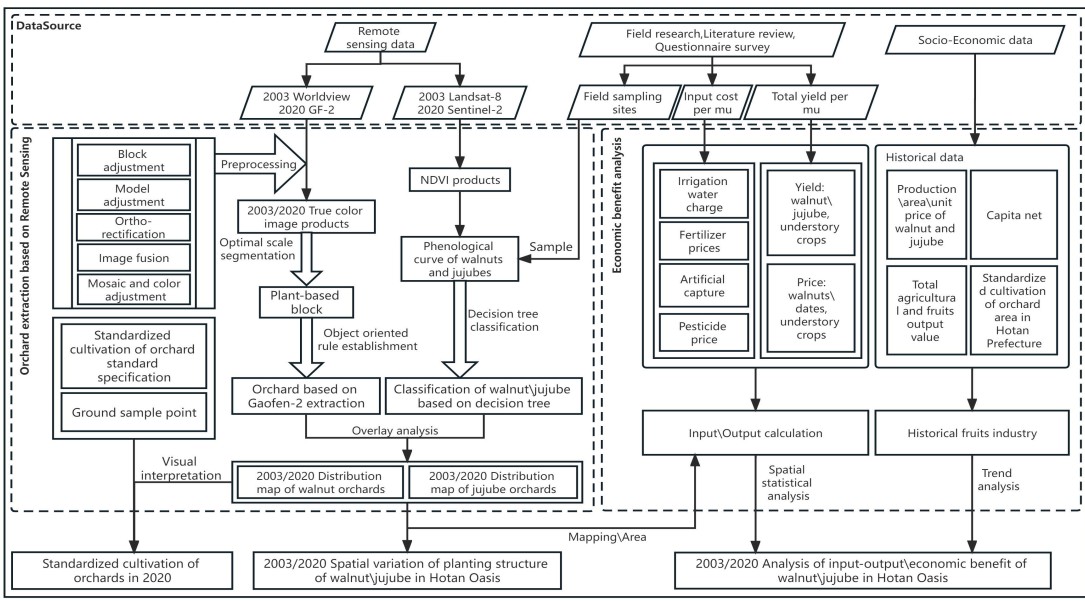

**Figure 2.** Technical flowchart of this study.

### 2.3.2. Satellite Image Preparation

The GF-2/WV-2 images were preprocessed using digital orthophoto production technology, including block adjustment, model adjustment, ortho-rectification, image fusion, mosaicking, and colour adjustment. The Sentinel-2 Atmospheric Reflective Top (L1C) products were subjected to atmospheric correction in Sen2Cor_v2.9 software, and all bands were subsequently resampled to a 10 m resolution on the Sentinel application platform (SNAP) for fusion purposes. The Landsat 5 TM C01/T1_SR datasets derived from the GEE were geometrically and atmospherically corrected using the Landsat Disturbance Adaptive Processing System.

### 2.3.3. Extraction of Walnut and Jujube Orchards

Based on the object-based information analysis method, the high-resolution images (WV-2 and GF-2) were first segmented using a multiresolution segmentation algorithm in eCognition Developer 8.0 software. This includes three main segmentation parameters: (1) band weights for segmentation, (2) scale parameters, and (3) brightness/shape and smoothness/compactness weights. In this work, all bands were used in the segmentation process, so a band weight of 1 was assigned. According to the spectral and texture characteristics of orchards and farmlands during the different seasons, the segmentation scale

parameters differed between the spring and winter and between the summer and autumn. Fruit trees did not grow leaves or bear fruit in spring and winter, so the segmentation scale was set to 180 to obtain relatively complete patches. In the summer and autumn, the fruit trees were already leafy and bore fruit, so the segmentation scale was increased to 200 to facilitate tree canopy separation from the understorey crops. A weight shape of 0.6 was used, whereas the compactness received a weight of 0.5. Second, a rule set for land classification was established by combining texture features, brightness features, and NDVI, while intercropping orchards, monocropping orchards, farmlands, and other areas (settlements, water bodies, unused land, etc.) could be accurately classified. Third, fifty training patches of walnut and jujube orchards were selected, and the NDVI time series curves of these two fruit crops were extracted from multitemporal Landsat 5 TM and Sentinel-2 images. With the use of the phenological features of walnut and jujube, the spectral thresholds of the critical period of walnut and jujube identification were calculated, and a decision tree model for walnut and jujube classification was constructed. Finally, the classification results were overlaid on the extracted orchard patches (including monocropping and intercropping orchards) based on high-resolution remote sensing images, and a spatial distribution map of walnut and jujube orchards in the Hotan Oasis was obtained.

In the field survey, we also found that the local government rapidly developed more standardized orchard cultivation areas to improve the quality and efficiency of the fruit industry. Standardized cultivation of walnut orchards exhibited a higher density (3 m × 3 m or 3 m × 5 m), while the traditionally planted walnut orchards exhibited a lower density (4 m × 5 m or 6 m × 6 m). Standardized cultivation of jujube orchards exhibited a density of 2 m × 3 m during the entire fruit period. Based on the location information of 28 local standardized walnut/jujube orchards provided by experts at the Hotan Forestry Bureau, we established interpretation rules for standardized walnut/jujube orchards in high-resolution images according to texture features. Combined with expert knowledge interpretation, we identified standardized cultivation areas of walnut/jujube orchards from the orchard maps produced in the previous step. Raster maps of walnut/jujube orchards and standardized cultivation areas of orchards were converted into vector maps, and the areas of orchards were calculated in ArcGIS 10.0 software.

### 2.3.4. Accuracy Evaluation

The final accuracy of the walnut/jujube orchard maps was evaluated using a confusion matrix, including the overall accuracy (OA), user accuracy (UA), and producer accuracy (PA). A total of 755 field survey samples were selected, including 424 samples of intercropping walnut orchards, 263 samples of jujube orchards, and 68 samples of monocropping walnut orchards. As indicated in Table 3, the OA value of the classification results was 96.82%. Among them, the UA and PA values of intercropping walnut and jujube orchards exceeded 95%, while both the UA and PA values of monocropping walnut orchards reached approximately 90%. This indicated that the results were reliable and could meet the precision requirements of remote sensing monitoring of forest fruit crops. However, the monocropping walnut orchards with small plot areas were easily confused with intercropping walnut and jujube orchards.

**Table 3.** Accuracy evaluation of orchard extraction in 2020.

| Post-Classification | Reference | | | Producer Accuracy/% | User Accuracy/% |
|---|---|---|---|---|---|
| | Intercropping Walnut | Jujube | Monocropping Walnut | | |
| Intercropping walnut | 422 | 0 | 4 | 98.6 | 98.14 |
| Jujube | 1 | 263 | 0 | 99 | 97.41 |
| Monocropping walnut | 1 | 0 | 64 | 90.14 | 92.75 |
| Overall accuracy: 96.82%; kappa coefficient: 98.55% | | | | | |

2.3.5. Economic Benefit Analysis Method

The economic benefit analysis includes a statistical analysis and a cost/output value analysis, which entails static statistical analysis. The basic unit of statistical analysis was one hectare (ha). Walnut/jujube trees and crops under the fruit trees are the critical components of the agroforestry system in this study. Due to the local government's poverty alleviation policy for walnut tree growers, the government subsidizes the intercropping mode, and there is no cost for orchard construction at the early stages. Regarding the monocropping mode, we considered the economic benefits of the fruit trees over five years after planting, so the cost of garden construction was also not included. The economic inputs included labour, irrigation, fertilizer, pesticides, and agricultural machinery. The economic output value included fruits and the crops under the fruit trees. All input and output prices are market prices for 2003 and 2020. The economic benefits analysis focuses on the structure of the input and output and the ratio of the output to the input.

## 3. Results

### 3.1. Spatiotemporal Change in Walnut and Jujube Orchards in the Hotan Oasis from 2003 to 2020

In 2003, the areas of walnut and jujube orchards in the Hotan Oasis were $18.05 \times 10^3$ ha and $19.1 \times 10^2$ ha, respectively (Figure 3a). Walnut orchards were mainly distributed in the upper parts of the alluvial plain oasis (UAPO) between the Karakash River and the Yurungkash River. The walnut orchard area in Hotan County ($61.8 \times 10^2$ ha) was the largest, followed by Luopu County ($58.9 \times 10^2$ ha), Moyu County ($52.9 \times 10^2$ ha), and Hotan city ($6.9 \times 10^2$ ha). The first three counties accounted for 96.18% of the total walnut orchard area in the Hotan Oasis. Jujube orchards were mainly distributed along the upper and middle reaches of the Karakash River. More than 80% of the jujube orchards in the Hotan Oasis were concentrated in Hotan County and Luopu County, with areas of 820 and 740 ha, respectively. Moyu County contained the smallest area of jujube orchards, at only 120 ha.

In 2020, the walnut and jujube orchard areas increased to $40.23 \times 10^3$ ha and $33.6 \times 10^3$ ha, respectively. The walnut orchards expanded to the lower parts of the alluvial plain oasis (LAPO) between the east bank of the Yurungkash River and the west bank of the Karakash River. The area of walnut orchards in Moyu County ($13.40 \times 10^3$ ha) was the largest, and that in Hotan city ($63.4 \times 10^2$ ha) was still the smallest. The proportion of the total walnut orchard area in the three primary walnut-producing counties (Moyu County, Hotan County, and Luopu County) was reduced to 84.24%. Jujube orchards were mainly distributed in the alluvial plain oasis between the two rivers and the newly reclaimed oasis in western Moyu County and eastern Luopu County. Some large areas of jujube orchards occurred in the downstream oasis and desert ecotone, and even in the sandy desert region. The main producing area of jujube orchards in the Hotan Oasis still occurred in Hotan County and Luopu County, with areas of $12.02 \times 10^3$ ha and $10.42 \times 10^3$ ha, respectively. The jujube orchards in these two counties accounted for 66.79% of the total jujube orchard area in the Hotan Oasis. The area of jujube orchards in Hotan city was the smallest, with an area of $27.6 \times 10^2$ ha (Figure 3b).

The area of walnut orchards in the Hotan Oasis continued to increase from $18.05 \times 10^3$ ha in 2003 to $40.23 \times 10^3$ ha in 2020. The increased walnut orchards were primarily located outside the old walnut orchards of the UAPO and in the newly reclaimed oasis of the LAPO. They also extended to the eastern bank of the Yurungkash River and the western bank of the Karakash River. The walnut orchard area increased the most in Moyu County and the least in Luopu County. Although the walnut orchard area in Hotan city was still the smallest, it significantly increased (8.19 times) from 2003 to 2020 (Figure 3c). The areas of jujube orchards also dramatically increased from $19.1 \times 10^2$ ha in 2003 to $33.6 \times 10^3$ ha in 2020. The expanded areas of jujube orchards were mainly situated near the edge of the desert in Luopu County and Moyu County, as well as the lower reaches of rivers in Hotan County. Hotan County exhibited the most significant increase in the jujube orchard area, while the smallest increase was observed in Hotan city. The old oasis of Hotan

provides suitable water and heat conditions, superior irrigation conditions, relatively fertile soil, and a relatively humid climate. However, there are many mature walnut orchards, and there is no space for the development of jujube orchards. Jujube trees are more light tolerant, drought resistant, cold resistant, and barren resistant than walnut trees. The newly reclaimed jujube orchards were basically located at the edge of the desert in Luopu County and Moyu County. The water resources in this region are relatively scarce, and the climate is dry, which is conducive to the control of jujube tree diseases and insect pests. Moreover, the jujube trees in this area constitute an important part of the shelterbelts occurring at the edge of the oasis and provide higher economic benefits than other ecological protection forests. Therefore, the local government encourages enterprises and farmers to reclaim land for planting jujube trees at the edge of the desert in Luopu and Moyu counties. Over the past 20 years, the area expansions of jujube orchards were much greater than those of walnut orchards. Except for Hotan city, which is limited by the cultivated area, the remaining three counties exhibited vigorous development of walnut and jujube orchards, and the development speed was more balanced (Figure 3d).

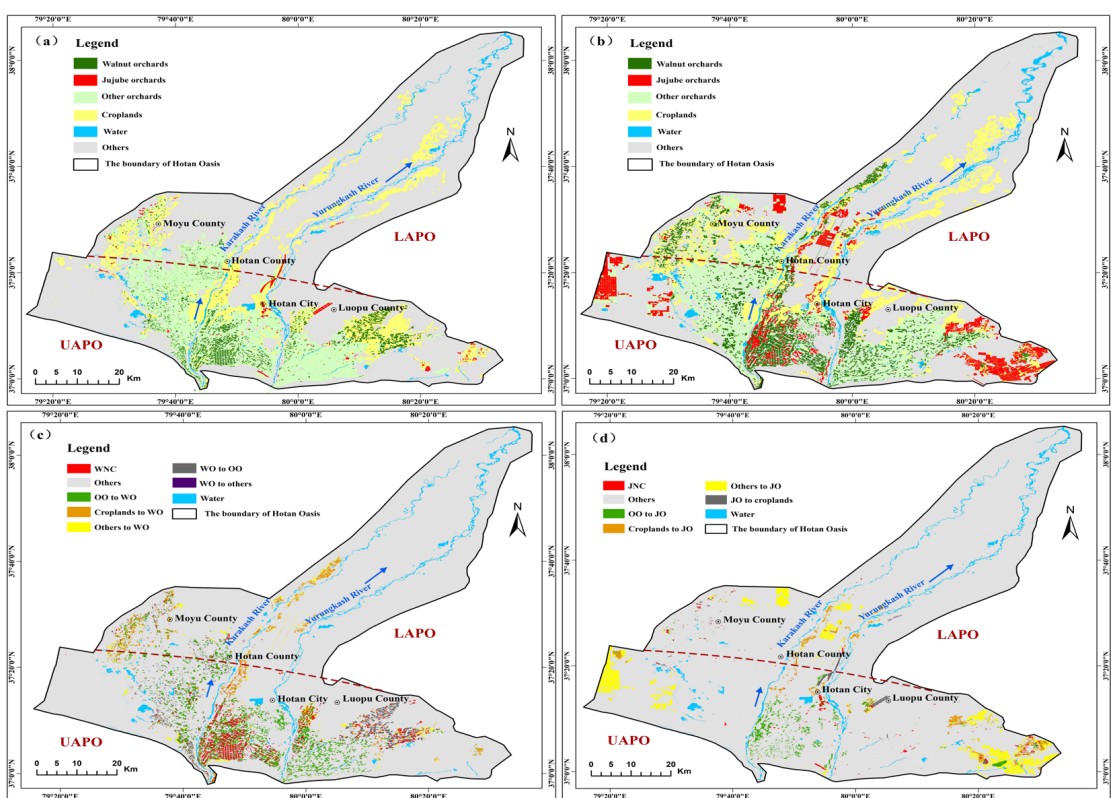

**Figure 3.** Maps of walnut/jujube orchards ((**a**) 2003, (**b**) 2020), and the spatiotemporal variation in walnut/jujube orchards between 2003 and 2020 ((**c**) walnut orchards, (**d**) jujube orchards). Notes: WNC denotes a walnut orchard, no change; OO denotes another type of orchard; WO denotes a walnut orchard; JNC denotes a jujube orchard, no change; JO denotes a jujube orchard; LAPO is the lower part of the alluvial plain oasis; UAPO is the upper part of the alluvial plain oasis; The blue arrows indicate the direction of the river; The red line represents the dividing line between UAPO and LAPO.

From 2003 to 2020, the increased areas of walnut orchards in the Hotan Oasis were mainly attributed to the conversion of other orchards ($21 \times 10^3$ ha) and croplands ($87.4 \times 10^2$ ha), while the decreased areas were mainly converted into other orchards ($71.7 \times 10^2$ ha) and jujube orchards ($21.1 \times 10^2$ ha) (Figure 4a). The areal conversion trend in walnut orchards in each county was consistent with that in the Hotan Oasis. Accounting for a significant proportion of the walnut area in Moyu County, the increased walnut orchard area

mainly originated from other orchards ($67.9 \times 10^2$ ha) and farmlands ($40.76 \times 10^2$ ha). The expanded areas of jujube orchards mainly originated from other land areas ($19.76 \times 10^3$ ha), croplands ($57.7 \times 10^2$ ha), and other orchards ($53.6 \times 10^2$ ha). In contrast, the transferred land involved cropland ($16.6 \times 10^2$ ha) (Figure 4a). Accounting for a significant proportion of the jujube area in Luopu County, the increased jujube orchard area mainly originated from other land ($82.51 \times 10^2$ ha) and cropland areas ($25.09 \times 10^2$ ha).

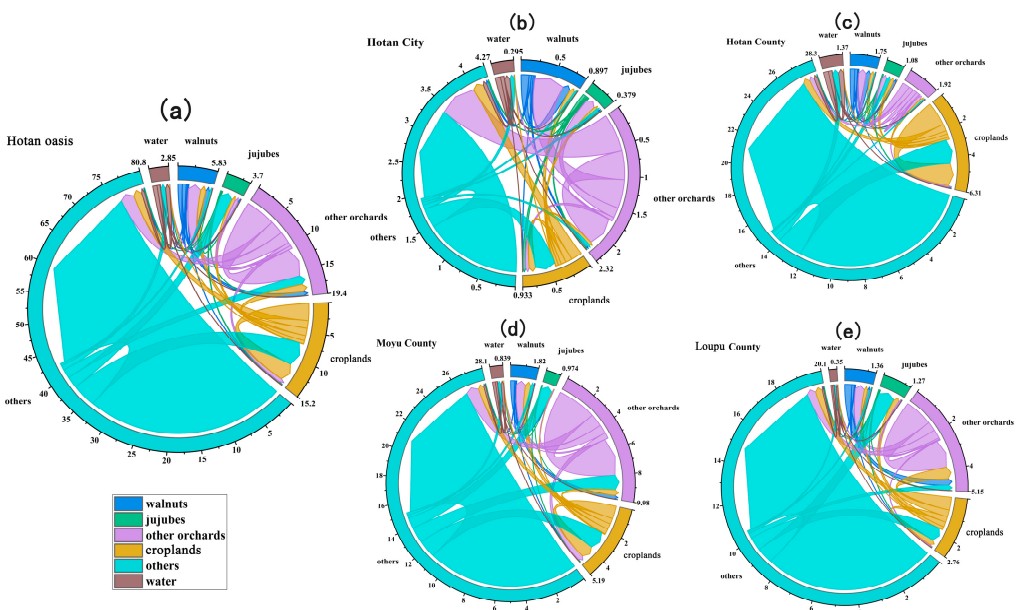

**Figure 4.** Land transformation of walnut/jujube orchards from 2003 to 2020 in the Hotan Oasis (**a**), Hotan city (**b**), Hotan County (**c**), Moyu County (**d**), and Luopu County (**e**).

### 3.2. Spatial Distribution of Standardized Cultivation in Orchards

Standardized cultivation of walnut orchards mainly entailed conversion from traditionally planted walnut orchards in the upper reaches of the Karakash River and the Yurungkash River. Standardized cultivation of jujube orchards always comprised newly cultivated orchards built in the lower reaches of the oasis and desert ecotone (Figure 5). The total area of standardized cultivation of walnut/jujube orchards was $24.07 \times 10^3$ ha in the Hotan Oasis, accounting for 90.24% of the total standardized cultivation area of orchards in Hotan Prefecture.

By 2020, the standardized cultivation areas of walnut orchards in Hotan County were the largest ($11.2 \times 10^2$ ha), while the areas in Luopu County were the smallest (280 ha) (Figure 6a). Standardized cultivation of jujube orchards was mainly distributed in Luopu County and Moyu County, with planting areas of up to $96.7 \times 10^2$ ha and $87.1 \times 10^2$ ha, respectively. The planting area in Hotan city was the smallest at only 20 ha (Figure 6b). In 2020, the standardized cultivation of walnut/jujube orchards in the Hotan Oasis accounted for 32.6% of the total area of traditionally planted orchards. Among them, the area ratio of orchards with standardized cultivation to traditionally planted orchards was the highest in Luopu County (49.61%) and the lowest in Hotan city (5.73%). The standardized cultivation area of walnut orchards accounted for 7.07% of the total area of traditionally planted orchards, and the standardized cultivation area of jujube orchards accounted for 63.26% of the total area. Hotan County and Hotan city exhibited higher ratios of the standardized cultivation area of walnut orchards to the area of traditionally planted walnut orchards at 9.65% and 7.72%, respectively. Luopu County exhibited the lowest area ratio of 3.43% (Figure 6c). The ratio of the standardized cultivation area of jujube orchards to the area of traditionally planted jujube orchards was the highest in Moyu County, reaching 95.86%, and it was the lowest in Hotan city at only 0.76% (Figure 6d).

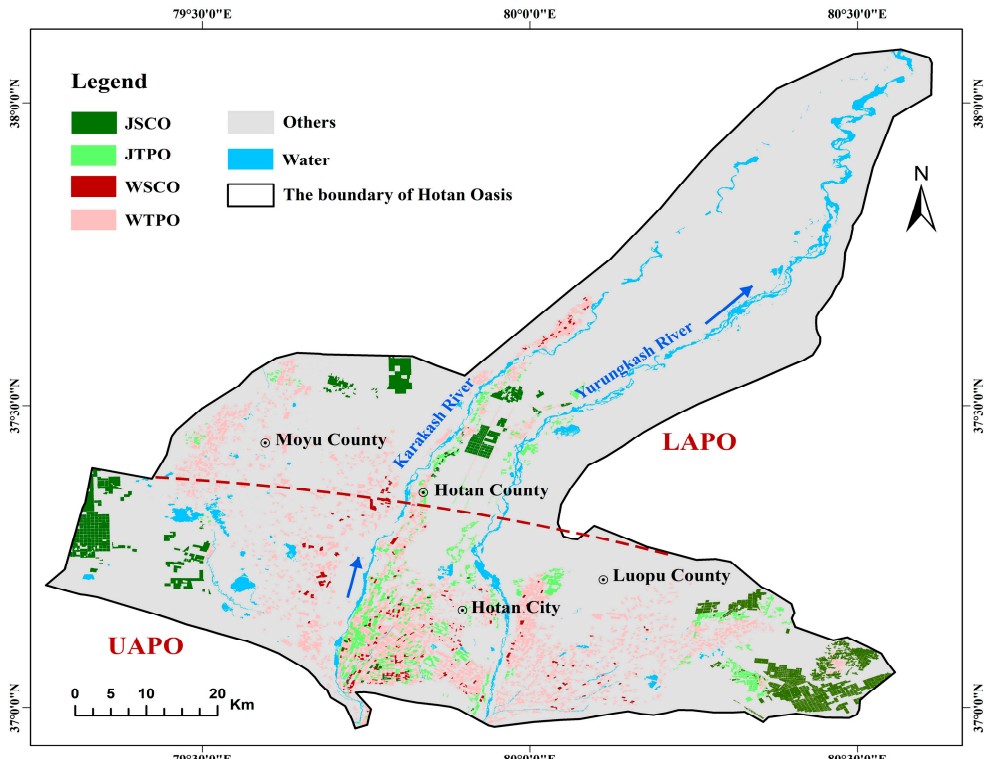

**Figure 5.** Map of standardized cultivation of walnut/jujube orchards in the Hotan Oasis in 2020. Notes: WSCO denotes standardized cultivation of walnut orchards; JSCO denotes standardized cultivation of jujube orchards; WTPO denotes a traditionally planted walnut orchard; JTPO denotes a traditionally planted jujube orchard; LAPO is the lower part of the alluvial plain oasis; UAPO is the upper part of the alluvial plain oasis. The blue arrows indicate the direction of the river; The red line represents the dividing line between UAPO and LAPO.

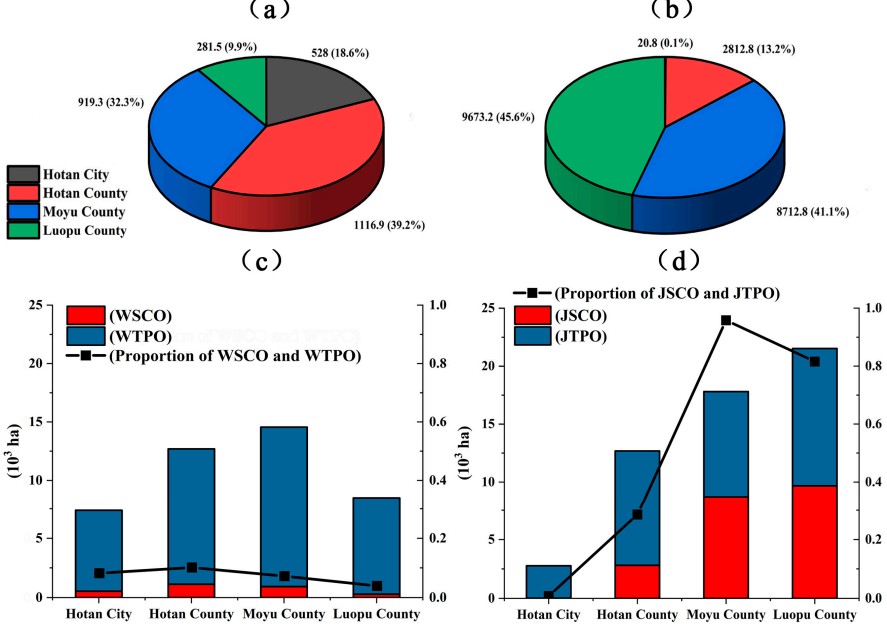

**Figure 6.** Proportion of the standardized cultivation area of walnut (WSCO) (**a**)/jujube (JSCO) (**b**) orchards for each country in the Hotan Oasis; area ratio of WSCO (**c**)/JSCO (**d**) orchards to traditionally planted walnut (WTPO)/jujube (JTPO) orchards in the Hotan Oasis.

### 3.3. Economic Benefit Evaluation of Walnut and Jujube Orchards

### 3.3.1. Economic Benefit Evaluation of Walnut Orchards

To increase the income of fruit growers and stabilize food production, winter wheat and corn were mainly planted under walnut trees in rotation. In 2003, the total cost per ha of traditionally planted walnut orchards was CNY 5850, of which the highest cost was observed for fertilizer (CNY 2475), followed by labour (CNY 1500) and irrigation costs (CNY 1125). The lowest costs were associated with machinery (CNY 600) and pesticides (CNY 150) (Table 4). However, in 2020, the total costs per ha of traditionally planted walnut orchards reached CNY 10,500, among which the highest cost was observed for fertilizer (CNY 3750), the costs of labour and irrigation increased to CNY 3000 and CNY 2400, respectively, and the costs of machinery and pesticides increased to CNY 1050 and CNY 300, respectively. The cost of traditionally planted walnut orchards in 2020 nearly doubled the 2013 levels, with the highest cost increase stemming from labour (accounting for 32% of the total increase) and the lowest cost increase resulting from machinery (accounting for 9.7% of the total increase). It was also found that the cost of labour was much higher than that of machinery (such as irrigation and fertilizer equipment). Since 2015, the Hotan government has made great efforts to renovate medium- and low-yield orchards and has promoted the construction of orchards involving standardized cultivation for obtaining high yields. Compared with the traditionally planted orchards in 2020, the total cost per ha of orchards with standardized cultivation reached CNY 21,450, with higher labour (CNY 7500), organic fertilizer (CNY 6750), and chemical fertilizer costs (CNY 3000). However, the cost of standardized cultivation in orchards increased more notably than that of traditionally planted orchards.

**Table 4.** Comparison of the economic cost and output value of traditionally planted and standardized cultivation-based walnut orchards in 2003 and 2020.

| | Year | 2003 | 2020 | |
|---|---|---|---|---|
| | Type | Traditionally planted walnut orchards | Traditionally planted walnut orchards | Standardized cultivation of walnut orchards |
| Cost | Irrigation per ha (CNY) | 1125 | 2400 | 2400 |
| | Chemical fertilizer per ha (CNY) | 975 | 1500 | 3000 |
| | Organic fertilizer per ha (CNY) | 1500 | 2250 | 6750 |
| | Pesticide per ha (CNY) | 150 | 300 | 600 |
| | Labour per ha (CNY) | 1500 | 3000 | 7500 |
| | Machinery per ha (CNY) | 600 | 1050 | 1200 |
| | Total input per ha (CNY) | 5850 | 10,500 | 21,450 |
| Output value | Walnut — Yield per ha (kg) | 1500 | 3000 | 6000 |
| | Walnut — Purchase price per kg (CNY) | 25 | 10 | 12 |
| | Walnut — Output value per ha (CNY) | 37,500 | 30,000 | 72,000 |
| | Winter wheat — Output value per ha (CNY) | 1200 | 5022 | 5022 |
| | Field corn — Output value per ha (CNY) | 900 | 1500 | 1500 |
| | Gross output value per ha (CNY) | 39,600 | 36,522 | 78,522 |
| | Net output value per ha (CNY) | 33,750 | 26,022 | 57,072 |
| | Total orchard area (ha) | 18,054.13 | 39,560 | 20,000 |
| | Total walnut production ($10^5$ tons) | 0.27 | 1.19 | 1.20 |
| | Total gross output value (CNY $10^8$) | 7.15 | 14.45 | 15.7 |
| | Total net output value (CNY $10^8$) | 6.09 | 10.29 | 11.41 |
| | Output/input ratios (%) | 676.92% | 347.83% | 366.07% |

Note: The above data are all obtained from questionnaires and statistical yearbooks.

The total output value of traditionally planted walnut orchards with cropping agro-forestry includes walnuts, winter wheat, and field corn (including straw) under the trees.

From 2003 to 2020, the net output value per ha of traditionally planted walnut orchards decreased from CNY 39,600 to CNY 36,522. However, the cost of walnut orchards involving standardized cultivation increased by nearly double to CNY 78,522. With the continuous increase in walnut orchard areas and yields over nearly 20 years, the gross output value of traditionally planted walnut orchards increased from CNY $71.5 \times 10^7$ in 2003 to CNY $14.45 \times 10^8$ in 2020. Moreover, the net output value increased from CNY $60.9 \times 10^7$ to CNY $10.29 \times 10^8$. Although the cost of walnut orchards increased by approximately two times, the purchase price of walnuts decreased by nearly half from 2003 to 2020. However, the walnut yield per ha increased by approximately six times, resulting in the net income of walnut orchards per ha not decreasing. The total net output value increased by approximately 1.7 times with an increase walnut orchard area in the Hotan Oasis. The construction of orchards involving standardized cultivation improved the quality and yield of walnuts. The yield of walnuts also increased by 15%–20% relative to traditionally planted orchards, and the purchase price of walnuts was 10% higher than that of walnuts cultivated in traditionally planted orchards. Therefore, the ratio of the output value to the cost of standardized cultivation in orchards per ha also increased by 1.1 times relative to traditionally planted orchards in 2020 (Table 4).

In 2003, more than 95% of the net output value of walnuts in the Hotan Oasis was derived from Hotan County, Luopu County, and Moyu County, at CNY $14.5 \times 10^7$, CNY $13.8 \times 10^7$, and CNY $12.4 \times 10^7$, respectively (Figure 7a). Moreover, the net output value in Hotan County was the lowest, at only CNY $16 \times 10^6$. In 2020, Moyu County became the county with the most significant proportion of the net output value of walnuts in the Hotan Oasis (CNY $36.3 \times 10^7$), followed by Hotan County (CNY $32.1 \times 10^7$) and Luopu County (CNY $21.6 \times 10^7$) (Figure 7b). Although the net output value of walnuts in Hotan city still accounted for the lowest proportion among all cities in the Hotan Oasis (accounting for 16.04% of the total value), it increased by 9.75 times. From 2003 to 2020, the total net output value of walnuts in the Hotan Oasis increased by 1.53 times, with the largest increase in Moyu County and the smallest increase in Luopu County.

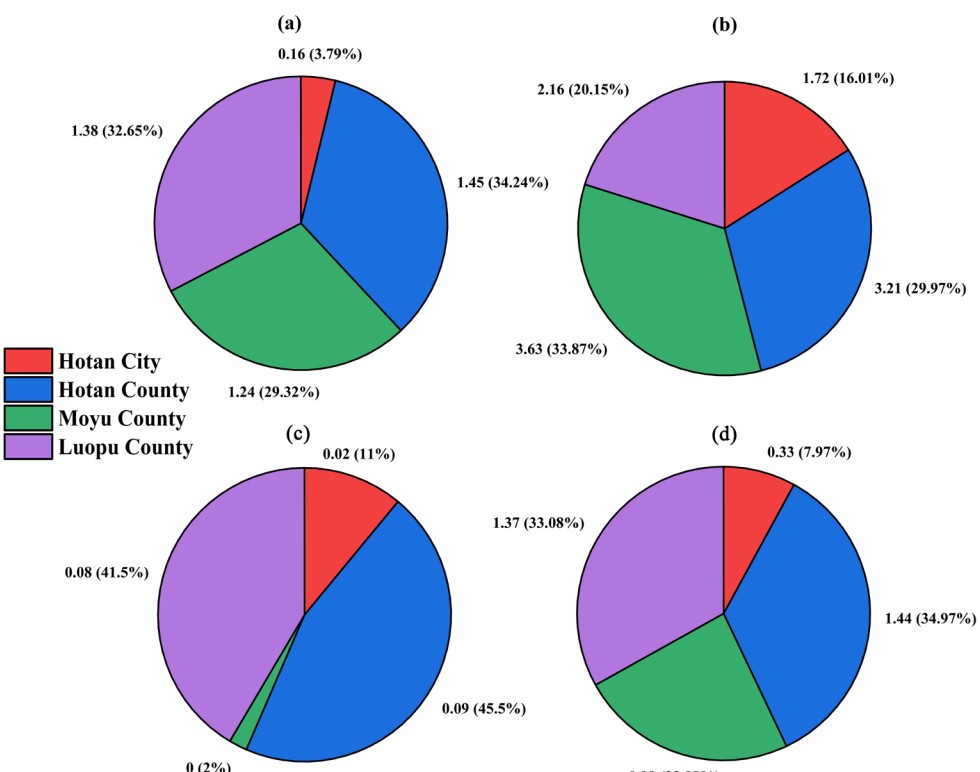

**Figure 7.** Net output value composition of walnuts (**a**,**b**) and jujubes (**c**,**d**) in the Hotan Oasis in 2003 and 2020.

### 3.3.2. Economic Benefit Evaluation of Jujube Orchards

Based on the field investigation, most jujube orchards adopted the cultivation model of dwarfing dense planting, and no other crop was planted under the trees to ensure excellent air and light conditions between the trees. In 2003, the total cost per ha of traditionally planted jujube orchards reached CNY 8850, of which the highest cost was observed for labour (CNY 4500), followed by fertilizer (CNY 2400) and irrigation (CNY 900), and the lowest costs were obtained for pesticides (CNY 750) and machinery (CNY 300) (Table 5). In 2020, the total cost per ha of traditionally planted jujube orchards reached CNY 22,950, approximately 2.3 times higher than that in 2003. The largest increase in costs was observed for organic fertilizer and labour, and the smallest increase was observed for machinery and irrigation. In 2020, the total increase in the cost per ha of standardized cultivation in jujube orchards was CNY 11,400, which is higher than that of traditionally planted orchards. Higher costs were observed for labour (CNY 12,000) and organic fertilizer (CNY 8400), with a lower cost for machinery (CNY 900). The yield per ha of traditionally planted jujube orchards increased from 600 kg in 2003 to $41.48 \times 10^2$ kg in 2020, but that of standardized cultivation-based orchards reached $75 \times 10^2$ kg in 2020.

**Table 5.** Comparison of the economic cost and output value of traditionally planted and standardized cultivation-based jujube orchards in 2003 and 2020.

| | Year | 2003 | 2020 | |
| --- | --- | --- | --- | --- |
| | Type | Traditionally planted orchards | Traditionally planted orchards | Standardized cultivation orchards |
| Cost | Irrigation per ha (CNY) | 900 | 1800 | 1800 |
| | Chemical fertilizer per ha (CNY) | 1200 | 4200 | 7500 |
| | Organic fertilizer per ha (CNY) | 1200 | 4800 | 8400 |
| | Pesticide per ha (CNY) | 750 | 3750 | 3750 |
| | Labour per ha (CNY) | 4500 | 7500 | 12,000 |
| | Machinery per ha (CNY) | 300 | 900 | 900 |
| | Total costs per ha (CNY) | 8850 | 22,950 | 34,350 |
| Output value | Jujube — Yields per ha (kg) | 600 | 4147.5 | 9000 |
| | Jujube — Purchase price per kg (CNY) | 40 | 10 | 12 |
| | Gross output value per ha (CNY) | 24,000 | 41,475 | 108,000 |
| | Net output value per ha (CNY) | 15,150 | 18,525 | 73,650 |
| | Total orchards area (ha) | 1798 | 35,525.73 | 3333.33 |
| | Total jujube yield ($10^5$ tons) | 0.01 | 1.47 | 0.3 |
| | Total gross output value (CNY $10^8$) | 0.43 | 14.73 | 3 |
| | Total net output value (CNY $10^8$) | 0.27 | 6.58 | 1.85 |
| | Output/input ratios (%) | 271.19% | 180.72% | 214.41% |

Note: The above data were all obtained from questionnaires and the statistical yearbook.

Although the purchase price of jujubes significantly decreased by 75% from 2003 to 2020, the total gross output value and the net output value of traditionally planted jujube orchards increased by 33.26 and 23.37 times, respectively, due to the 155-fold increase in the total jujube yield. Against the background of low prices on the domestic jujube market, the net output value per ha of standardized cultivation-based jujube orchards with higher yields and quality levels reached CNY 73,650, which was 2.98 times higher than that of traditionally planted orchards. As the area of standardized cultivation-based jujube orchards accounted for 9.38% of that of traditionally planted jujube orchards, the total net output value reached 28.12%. The ratio of the output value to the cost per ha of standardized cultivation-based jujube orchards was 18.64% higher than that of traditionally planted orchards.

In 2003, Hotan County and Luopu County jointly accounted for 87.63% of the total net output value of jujubes in the Hotan Oasis, at CNY $9 \times 10^6$ and CNY $8 \times 10^6$, respectively

(Figure 7c). Moyu County exhibited the lowest net output value of jujubes (CNY $4 \times 10^5$), accounting for 2.06% of the total value. In 2020, the proportion of the jujube revenue in Hotan County and Luopu County decreased to 68.04%, but that in Moyu County increased to 23.97% (Figure 7d). Hotan city exhibited the highest decrease in revenue stemming from jujubes, accounting for 7.99% of the total value. From 2003 to 2020, the total net output value of jujubes in the Hotan Oasis significantly increased from CNY $19.4 \times 10^6$ to CNY $41.3 \times 10^7$, representing an increase of 20.29 times. Hotan County and Luopu County contributed the most to the growth in jujube income, while Hotan city exhibited the least growth.

## 4. Discussion

### 4.1. Comparison of the Accuracy of the Fruit Extraction Results

In this study, we first used multiscale segmentation and object-oriented methods based on high-resolution images to extract the high-precision boundaries of orchard patches. Then, the types of orchard patches were classified using a decision tree model, which was built using field sample points and NDVI time series curves of multitemporal images. We combined the high spatial accuracy of the high-resolution images and the phenological curve integrity of multitemporal images. The overall accuracy of walnut and jujube orchard extraction was 96.82%. Although fruit tree resources are abundant in Xinjiang, few studies have been conducted considering fruit tree extraction using remote sensing. Local scholars have used multisource remote sensing images to extract the areas of jujube, apple, and pear orchards in the Tarim Basin in southern Xinjiang, and the overall classification accuracy was 88.43%, 78.14%, and 86.36%, respectively [21–23]. Comparing the classification accuracy of different orchards in similar areas, the overall accuracy of the planting areas of walnut and jujube orchards extracted in the Hotan Oasis reached more than 90%, which is relatively high. The most significant limitation of using multispectral data for extracting orchard types is the lack of accurate orchard boundary information. However, in this study, we used high-resolution images to compensate for this limitation to improve the overall extraction accuracy of orchards. In addition, most orchard information extraction studies focused on the monocropping pattern areas (e.g., citrus, pear, apple, hazelnut, and cashew) [11,24,25], which exhibit less surface disturbance information than intercropping orchards, so the extraction difficulty is lower.

### 4.2. Changes in the Production and Income of Walnut and Jujube Orchards in the Hotan Oasis over the Past 20 Years

Walnut trees planted in the Hotan Oasis experienced three stages of slow growth, rapid growth, and adjustment from 2003 to 2020 (Figure 8a–c). During the slow growth period (2003–2008), the area and yield of walnut orchards slowly increased, with average growth rates of $0.48 \times 10^3$ ha/year and $3.40 \times 10^3$ tons/year, respectively. Moreover, the purchase price per kg of walnuts in the Hotan Prefecture was approximately CNY 30, slightly higher than the national purchase price (Figure 9a). Therefore, the output value increased from CNY $0.78 \times 10^8$ to CNY $35.1 \times 10^7$, with a growth rate of $0.54 \times 10^8$ CNY/year. During the rapid growth period (2009–2017), the growth rates of the area and yield of walnut orchards reached $2.83 \times 10^3$ ha/year and $14.83 \times 10^3$ tons/year, 5.24 times and 4.36 times those at the previous stage, respectively. The early purchase price of walnuts in the Hotan Oasis was higher than the national average price, which encouraged farmers to expand the planting areas of walnut orchards. In addition, the local government adopted the fruit industry as the main focus of the local poverty alleviation efforts and vigorously developed fruit planting areas [26]. However, the purchase price per kilogram of walnuts decreased from CNY 26 in 2014 to a minimum value of CNY 13 in 2017. Especially in 2017, the purchase price per kilogram of walnuts decreased from an average value of approximately CNY 20 to CNY 13, resulting in a sudden reduction of 24.2% in the walnut output value compared with 2016 levels. Overall, the total income of walnuts during this period continued to increase, with an increase rate of $2.06 \times 10^8$ CNY/year. During the adjustment period

(2018–2020), the purchase price per kilogram of walnuts in China and Hotan Prefecture continued to decrease to CNY 10, which seriously affected the enthusiasm of farmers to plant walnuts. Since 2018, the local government has actively responded to the quality and efficiency improvement efforts of the forest and fruit industry in Xinjiang [27]. Many medium- and low-yield walnut orchards were converted into croplands or standardized cultivation-based orchards. The number of walnut orchards decreased by 12.12% in 2019 from the number in 2018, while the number of standardized cultivation-based orchards increased by 12.42%. Although the areas of walnut orchards decreased by $1.7 \times 10^3$ ha/year, the yields significantly increased by $17.59 \times 10^3$ tons/year, and the growth rate was higher than that at the previous two stages. Affected by the continued low purchase price of walnuts, the growth rate of the output value was $1.26 \times 10^8$ CNY/year, which is 38.83% lower than that during the rapid growth period.

Jujube planting in the Hotan Oasis also experienced three stages: slow growth, rapid growth, and adjustment from 2003 to 2020 (Figure 8a–c). During the slow growth period (2003–2008), the growth rates of the area and yield of jujube orchards were low, with average values of $0.32 \times 10^3$ ha/year and $2.27 \times 10^3$ tons/year, respectively. The average purchase price per kilogram of jujubes in the Hotan Prefecture was approximately CNY 33, which was 1.4 times the national price (Figure 9b). The output value increased from CNY $0.05 \times 10^8$ to CNY $0.81 \times 10^8$, with a growth rate of $0.16 \times 10^8$ CNY/year. During the rapid growth period (2009–2017), the area of jujube orchards increased from $5.16 \times 10^3$ ha in 2009 to $35.32 \times 10^3$ ha in 2017, with a high growth rate of $4.11 \times 10^3$ ha/year. The main reason for the expansion of jujube planting areas was that the purchase price of jujubes was higher than the national average price, as well as the strong support of the local government for the fruit industry. However, the jujube orchard yield and output value did not rapidly increase with the expansion. The yield increased from $12.59 \times 10^3$ tons to $83.39 \times 10^3$ tons, with a $7.7 \times 10^3$ tons/year growth rate. Although the purchase price per kilogram of jujubes was 2.1 times the national average (Figure 9b), it still decreased from CNY 45 in 2009 to CNY 10 in 2017. The output value of jujube orchards slightly increased from CNY $0.87 \times 10^8$ to CNY $33.2 \times 10^7$, with a growth rate of $33 \times 10^6$ CNY/year. During the adjustment period (2018–2020), the area of jujube orchards remained stable, and the growth rate reached only $0.09 \times 10^3$ ha/year. However, the jujube orchard yield and output value increased by $19.07 \times 10^3$ tons/year and $2.29 \times 10^8$ CNY/year, respectively, with the most significant growth rate among these three periods. Despite the continued decline in jujube prices on the domestic market, the local government did not blindly expand the areas of jujube orchards. The household investigation revealed that the planting technique of short, early, dense, and abundant jujube trees, widely promoted in recent years, greatly improved the yield and quality of jujubes in the Hotan Oasis.

Combining Figures 8 and 9, it was observed that the area, yield, and output value of walnuts and jujubes could be inconsistent with the increase in the purchase price. Although the purchase price of walnuts and jujubes has continued to decline over the past 20 years, the yield and output value have continued to grow. The reason is that over the past 20 years, the purchase price of walnuts and jujubes has been higher than the domestic purchase price, and local farmers have begun to expand the area of their orchards. As the area of orchards increased, the yield also increased. The increase in the yield inevitably resulted in a lower purchase price of these fruits. At the same time, lower prices affected farmers' incentives, so the planted area was not continuously expanded. However, the orchards that reached maturity still maintained high yields, and the fruit yield during the third phase continued to rise. Among them, the highest purchase price of jujubes occurred in 2011, after which the price continued to decline. However, the value of jujubes did not begin to rapidly increase until 2017. The reason is that the jujube yields steadily increased. To ensure the income of local farmers, jujubes should only be sold at a reduced price.

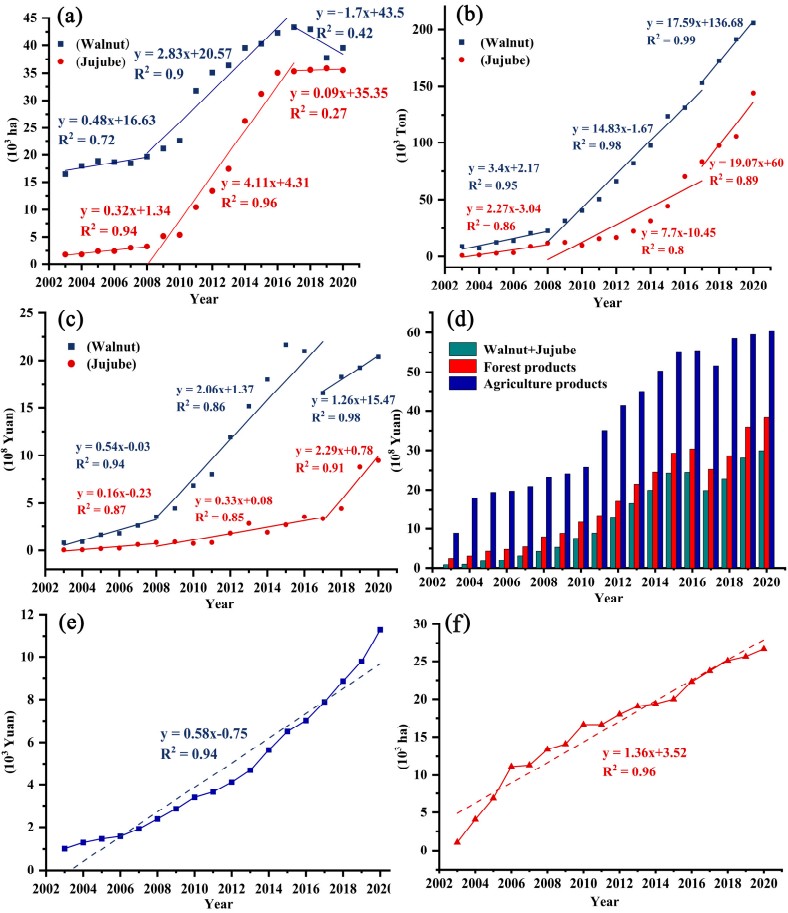

**Figure 8.** Production of walnut and jujube orchards (planting area (**a**), yield (**b**), and output value (**c**)), gross output value of agriculture and fruit products (**d**), annual net income of farmers (**e**), and area of standardized cultivation-based orchards (**f**) in the Hotan Oasis from 2003 to 2020.

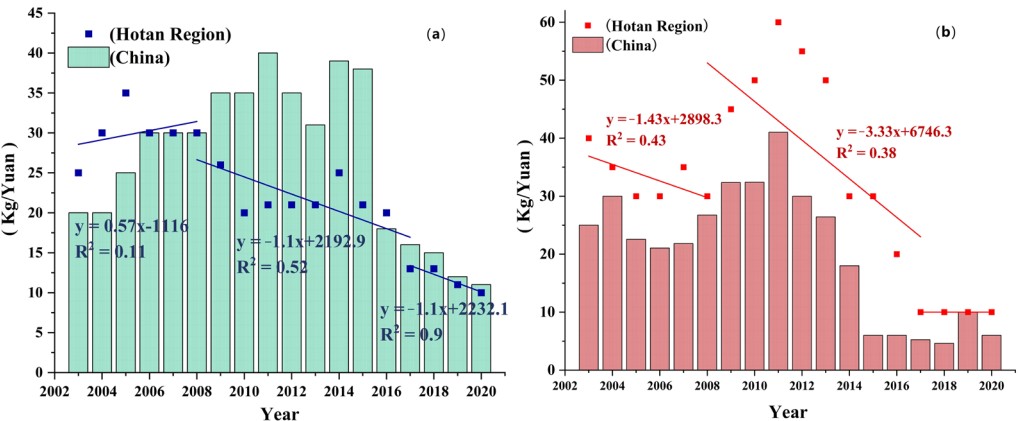

**Figure 9.** The purchase price per kilogram of walnuts (**a**) and jujubes (**b**) in China and the Hotan Region from 2003 to 2020.

The total output value of agriculture accounted for 50% of the gross national product of Hotan from 2016 to 2020 (Statistical Yearbook of Hotan District). During the slow development period of the forest fruit industry (2003–2008), the average gross output value of agriculture was CNY $18.33 \times 10^8$, and the proportion of the forest industry reached approximately 25.66%. During the rapid development period of the forestry and fruit industry (2009–2020), the gross output value of agriculture continuously increased from CNY $24.14 \times 10^8$ to CNY $60.5 \times 10^8$, while that of the forestry industry also increased

from CNY $8.84 \times 10^8$ to CNY $38.3 \times 10^8$. By 2020, the gross output value of forestry production accounted for 63.31% of the total agricultural production, while the output value of walnuts and jujubes accounted for 78.06% of the total forestry production value. The fruit industry, represented by walnuts and jujubes, has become the agricultural pillar of the Hotan Oasis. With the declining purchase price of walnuts and jujubes in China, the local government has vigorously promoted the quality and efficiency of the forestry and fruit industry. It has increased the construction of standardized cultivation-based orchards. The areas of standardized cultivation-based walnut/jujube orchards increased from $10.7 \times 10^2$ ha to $26.67 \times 10^3$ ha. The purchase prices of walnuts and jujubes grown in standardized cultivation-based orchards were 1.2 times higher than that of fruits grown in traditionally planted orchards. The income of local farmers continuously increased from CNY $10.2 \times 10^2$ in 2003 to CNY $11.29 \times 10^3$ in 2020. This showed that the forestry and fruit industry had promoted increasing the income of local farmers and realizing rural revitalization.

*4.3. Measures and Suggestions*

In Xinjiang, the fruit planting areas accounted for 25% of the total agricultural land area, fruit production accounted for 38% of the total crop production, and fruit income accounted for 54% of the total agricultural product income (Statistical Yearbook of Xinjiang, 2021). In addition, fruit products could be processed and transformed locally in a labour-intensive manner, primarily involving various industries, that are very suitable for southern Xinjiang with more people, less land, and fewer nearby employment opportunities. Therefore, the fruit industry is the leading agricultural industry in Xinjiang, while it is also an essential industrial support for poverty alleviation in the profoundly impoverished areas of southern Xinjiang.

Recently, the fruit industry in Xinjiang has faced the following problems: first, the fruit distribution structure is unclear, and the degree of homogenization is high [28]. Second, the level of orchard construction is uneven, and the management levels of standardized cultivation-based orchards, high-quality orchards, and traditionally planted orchards are very different. The rate of high-quality fruit per unit area is low [29]. Finally, the fruit tree area and production continued to significantly increase, but the increase in fruit farmer income was limited. We suggest implementing the following measures: (1) with the advantages of high-resolution remote sensing data, we should strengthen the monitoring and classification of growth, yields, etc., to accurately obtain the real-time situation of fruit tree cultivation throughout the region. From natural conditions and the planting history, we should fully explore and determine the characteristics of fruit products in each region, focusing on the development of one product in each county and one product in each village. In addition to planting regular types of fruits, we should introduce excellent varieties suitable for planting, and we must strengthen the National Geographic Indication product certification of fruits. (2) We need to enhance the transformation and construction of modern standard orchards. We should gradually phase out grain and cotton intercropping, implement a water distribution system based on the type of fruit, and promote the integrated supply of water and fertilizer. We should utilize the high yields and high-quality characteristics of standardized orchards to improve the competitiveness of fruit on the domestic market. (3) We propose promoting the model of leading enterprises + professional cooperatives + fruit farmers and building a modern fruit industry system.

## 5. Conclusions

Using high-resolution and multitemporal remote sensing data, we successfully extracted the distribution and area of walnut and jujube orchards in the fruit tree–crop intercropping system with an accuracy higher than 95%. From 2003 to 2020, the areas of jujube orchards were smaller than those of walnut orchards, but the expansion rate of jujube orchards was much higher than that of walnut orchards. The increased areas of walnut orchards were mainly attributed to the conversion of other types of orchards ($21 \times 10^3$ ha),

while those of jujube orchards mainly originated from other land use types ($19.76 \times 10^3$ ha). Therefore, the increased areas of walnut orchards were primarily located outside the old walnut orchards in the UAPO. However, jujube orchards were mainly situated in the LAPO area and desert ecotone. With the continuous promotion of modern orchard construction, standardized cultivation-based walnut/jujube orchards accounted for 32.6% of the total number of traditionally planted orchards by 2020. The economic benefits of walnut and jujube orchards, which were calculated based on accurate classification results derived from remote sensing, differed greatly. The positive benefits of area expansion and increased production were partially offset by the higher labour and irrigation costs, as well as a sharp drop in the purchase price, resulting in a 68.96% increase in the total net income of walnut orchards in 2020. Although the input per ha increased by 1.59 times and the purchase price per ha decreased by 75%, the area and yield increased by 18.75 times and 5.91 times, respectively, causing an increase in the total net income of jujubes of 23 times. The ratio of the output value to the cost of walnut orchards under the fruit tree–crop intercropping conditions was higher than that of jujube orchards. The intercropping mode in agroforestry provided suitable economic returns for farmers so that understory cash crops could be appropriately promoted in suitable orchards. Although the input per ha of standardized cultivation-based orchards was higher, the yield and purchase price were also higher, so the total net income was higher than that of traditionally planted orchards. This result indicated that it is necessary to vigorously promote standardized cultivation in orchards in the future to improve quality and quantity and thus achieve the goal of increasing production and income.

**Author Contributions:** Conceptualization, J.B. (Jie Bai) and A.B.; methodology, J.J. and J.L.; software, J.J.; validation, J.L. and H.H.; formal analysis, H.H.; investigation, J.J. and J.B. (Jiayu Bao); resources, A.B. and J.B. (Jie Bai); data curation, C.C.; writing—original draft preparation, J.J.; writing—review and editing, J.B. (Jie Bai); visualization, C.C. and J.B. (Jiayu Bao); supervision, J.B (Jie Bai); project administration, J.B. (Jie Bai) and A.B.; funding acquisition, J.B. (Jie Bai), A.B., C.C. and J.L. All authors have read and agreed to the published version of the manuscript.

**Funding:** This research was funded by the West Light Foundation of The Chinese Academy of Sciences (No. 2020-XBQNXZ-009), the Key Laboratory Project of Xinjiang Uygur Autonomous Region, China (No. 2020D04036), the Key R&D Program of Xinjiang Uygur Autonomous Region (Grant No. 2022B03021), the Tianshan Talent Training Program of Xinjiang Uygur Autonomous region (Grant No. 2022TSYCLJ0011), and the Tianshan Talent–Science and Technology Innovation Team (No. 2022TSYCTD0006).

**Data Availability Statement:** The data presented in this study are available on request from the corresponding author. The data are not publicly available due to that is the latest processed data and has not been checked and audited by relevant authorities.

**Acknowledgments:** We are grateful to those who participated in the data collection activities and data analysis, including Kader at the Xinjiang Academy of Forestry, and the younger students who participated in this study.

**Conflicts of Interest:** All the authors declare that they have no conflict of interest.

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
