# Peer review of "Cost–Benefit Evaluation of Walnut and Jujube Orchards under Fruit Tree–Crop Intercropping Conditions in Southern Xinjiang"

_forests, doi:10.3390/f14112259_

Round 1

Reviewer 1 Report

Comments and Suggestions for Authors

In this manuscript authors identified the spatial distributions, extents, and economic benefits of walnut and jujube orchards in Hotan Oasis of southern Xinjiang during 2003-2020. The manuscript is good written. Following are my concerns.

·  Why did the jujube orchards expand towards edge of the desert in Luopu County and Moyu County?

·  Did authors work on the varietal classification of jujube and walnut in those counties? Or did authors work on plants like grapes, apricot or other?

· Authors classified the jujube planting in Horan Oasis into three stages i.e. slow growth, rapid growth and adjustment from 2003 to 2020. However, its impact on purchase price per kilogram of jujube is unclear in fig 9(a).

·  Please explain the Normalized Difference Vegetation Index.

· Please clarify me and add necessary detail in manuscript too. Field survey of Hotan Oasis was carried out in late April 2021 following some questions. How authors calculated the socio-economic statistics during 2003 and 2020?

·  It is highly recommended to add the graphical abstract of whole experiment showing major steps of methodology and other activities.

·  There are many minor mistakes which need to be revised for example line 56-58, 67 and 132-134.

Comments on the Quality of English Language

Good, can be improved grammatically 

Author Response

Comments 1: Why did the jujube orchards expand towards edge of the desert in Luopu County and Moyu County?

Response 1: The old oasis of Hotan has good water and heat conditions, superior irrigation conditions, relatively fertile soil and relatively humid climate. However, there are many mature walnut orchards, and there is no space for the development of jujube orchards. As jujube tree is more light tolerant, drought resistant, cold resistant and barren resistance than walnut tree. The newly reclaimed jujube orchards are basically located in the edge of the desert in Luopu County and Moyu County. The water resources in this region are relatively scarce and the climate is dry, which is conducive to the control of jujube tree diseases and insect pests. Moreover, jujube tree here is also an important part of the shelterbelts on the edge of the oasis and has higher economic benefits than other ecological protection forests. Therefore, the local government encourages enterprises and farmers to reclaim land to plant jujube tree son the edge of the desert in Luopu and Moyu counties. And we've added this explanation to the conclusion section. (Seeing Lines: 357-368, paragraph 3.1)

Comments 2: Did authors work on the varietal classification of jujube and walnut in those counties? Or did authors work on plants like grapes, apricot or other?

Response 2: It is difficult to distinguish the varieties of fruit trees from remote sensing images, and we have not done this part of research at present. Since the walnut tree and jujube tree are the most important fruit tree types in Hotan oasis, we just extracted these two types of orchards in this study, and the classification of other orchards will be carried out in the future research.

Comments 3: Authors classified the jujube planting in Horan Oasis into three stages i.e. slow growth, rapid growth and adjustment from 2003 to 2020. However, its impact on purchase price per kilogram of jujube is unclear in fig 9(a).

Response 3: We have revised the Fig 9 to emphasize this point. (Seeing Lines: 651) in the section of Discussion. And we've added add a corresponding interpretive analysis in the discussion. (Seeing Lines: 601-615, paragraph 4.2)

Comments 4: Please explain the Normalized Difference Vegetation Index.

Response 4: The Normalized Difference Vegetation Index (NDVI) is a kind of index which can reflect the plants growth condition by combining the satellite detection data in different wavelength bands. The principle is that the leaf surface of plants has strong absorption characteristics in the visible red band and strong reflection characteristics in the near infrared band. The formula for NDVI is as follows: NDVI=(NIR-R)/(NIR+R). NIR represents the reflection value of the near-infrared band, and R represents the reflection value of the red band. The value interval: -1≤NDVI≤1. Negative values indicate that the ground is covered by cloud, water, snow, etc., which is highly reflective of visible light; 0 indicates rock or bare soil; the positive value indicates the presence of vegetation cover, which increase with increasing vegetation cover.

Comments 5: Please clarify me and add necessary detail in manuscript too. Field survey of Hotan Oasis was carried out in late April 2021 following some questions. How authors calculated the socio-economic statistics during 2003 and 2020?

Response 5: The data for the calculation of output-input ratio is based on the data from the field survey in 2021. The socio-economic data is used for the trend analysis of walnut and jujube industry for 2003-2020 are obtained by consulting statistical yearbooks and government bulletins, and it was used f. And we've added the necessary detail to the conclusion section. (Seeing Lines: 220-222, paragraph 2.3.1)

Comments 6: It is highly recommended to add the graphical abstract of whole experiment showing major steps of methodology and other activities.

Response 6: We have added the graphical abstract to emphasize this point. (Seeing Lines: 16-17).

Comments 7: There are many minor mistakes which need to be revised for example line 56-58, 67 and 132-134.

Response 7: We have revised these minor mistakes which need to be revised. (Seeing Lines: 58-61, Line: 69-72 in the section of Introduction and Line: 136-138 in the section of Materials and Methods.)

“As a significant fruit-producing region globally, China's fruit production reached 270 million tons, nearly accounting for half of the global fruit production (FAO Database 2019). Xinjiang is a critical fruit-producing area in China, with 13% of the national total fruit plantation area [6], and it is also one of the world's six fruit production regions.” has been revised to “As a significant fruit-producing region globally, China produces 270 million tons of fruit, accounting for nearly half of the global fruit production (FAO Database 2019). Xinjiang is a critical fruit-producing area in China, accounting for 13% of the national fruit plantation area [6], and it is also one of the world's six fruit-producing regions.” (Seeing Lines: 58-61)

“The income from fruit cultivation here accounts for more than 60% of local farmers' income and is the primary industry to increase the income of local farmers.” has been revised to “The income of fruit planting here accounts for more than 60% of local farmers' income, and the fruit industry is also one of the important economic industries to increase the income of local farmers.” (Seeing Lines: 69-72)

“Hotan Oasis is located in southern Xinjiang of Northwest China with an area of 3 110 km2, and it administratively comprises Hotan city, Hotan country, Moyu, and Luopu country.” has been revised to “Located in the southern part of Xinjiang in northwestern China, the Hotan Oasis covers an area of 3 110 km2 and has the administrative districts of Hotan city, Hotan county, Moyu county and Luopu county.” (Seeing Lines: 136-138)

4. Response to Comments on the Quality of English Language

Point 1:Good, can be improved grammatically

Response 1: Further refinements and modifications in gramma have been made.

Reviewer 2 Report

Comments and Suggestions for Authors

The authors present an interesting and comprehensive report of cultivars.

I agree with the need for improvement in standardization.

My suggestion is to include ways to better process crops, for example, use pectin to produce vitamin C with a simple fermentation process (10.3390/molecules25112706), or 10.1016/j.foodchem.2017.01.074 

Comments on the Quality of English Language

It needs some revision

Author Response

Comments 1: My suggestion is to include ways to better process crops, for example, use pectin to produce vitamin C with a simple fermentation process (10.3390/molecules25112706), or 10.1016/j.foodchem.2017.01.074

Response 1: This paper focuses on extracting the planting areas of walnut and jujube orchards using remote sensing data, and calculating the input cost and output benefit. The output benefits mainly refers to the purchase price of fruits, and the value of further processing of the fruit at a later stage does not considered. At present, there is no information in this regard, and the views of experts in this regard will be taken into account in future studies.

4. Response to Comments on the Quality of English Language

Point 1:It needs some revision

Response 1: Further refinements and modifications in language have been made.
